Preparation and characterization of gelatin-polysaccharide composite hydrogels for tissue engineering

Ye Jing 1
Yang Gang yang_gang@scu.edu.cn 1
Zhang Jing 1
Xiao Zhenghua 2
He Ling 1
Zhang Han 1
Liu Qi 1
1 College of Biomedical Engineering, Sichuan University , Chengdu , Sichuan , China
2 Department of Cardiovascular Surgery, West China Hospital of Sichuan University , Chengdu , Sichuan , China
Gonzalez-Ortega Omar
Electronic publication date: 2021 Mar 15
Publication date: 2021
Volume: 9
Electronic Location ID: e11022
Received 2020 Oct 15; Accepted 2021 Feb 7
Copyright: ©2021 Ye et al.
Copyright year: 2021
Copyright holder: Ye et al.
License: This is an open access article distributed under the terms of the Creative Commons Attribution License, which permits unrestricted use, distribution, reproduction and adaptation in any medium and for any purpose provided that it is properly attributed. For attribution, the original author(s), title, publication source (PeerJ) and either DOI or URL of the article must be cited.
License URL: https://creativecommons.org/licenses/by/4.0/

Keywords: Composite hydrogel, Gelatin, Chitosan, Alginate, Cardiac tissue engineering

Funding: The National Natural Science Foundation of China 61571314 Science and Technology Department of Sichuan Province, China 2018SZ0384 2020YFG0081 This work was financially supported by the National Natural Science Foundation of China (61571314) and the Science and Technology Department of Sichuan Province, China (2018SZ0384, 2020YFG0081). The funders had no role in study design, data collection and analysis, decision to publish, or preparation of the manuscript.

==============================
Background

Tissue engineering, which involves the selection of scaffold materials, presents a new therapeutic strategy for damaged tissues or organs. Scaffold design based on blends of proteins and polysaccharides, as mimicry of the native extracellular matrix, has recently become a valuable strategy for tissue engineering.

Objective

This study aimed to construct composite hydrogels based on natural polymers for tissue engineering.

Methods

Composite hydrogels based on blends of gelatin with a polysaccharide component (chitosan or alginate) were produced and subsequently enzyme crosslinked. The other three hydrogels, chitosan hydrogel, sodium alginate hydrogel, and microbial transglutaminase-crosslinked gelatin (mTG/GA) hydrogel were also prepared. All hydrogels were evaluated for in vitro degradation property, swelling capacity, and mechanical property. Rat adipose-derived stromal stem cells (ADSCs) were isolated and seeded on (or embedded into) the above-mentioned hydrogels. The morphological features of ADSCs were observed and recorded. The effects of the hydrogels on ADSC survival and adhesion were investigated by immunofluorescence staining. Cell proliferation was tested by thiazolyl blue tetrazolium bromide (MTT) assay.

Results

Cell viability assay results showed that the five hydrogels are not cytotoxic. The mTG/GA and its composite hydrogels showed higher compressive moduli than the single-component chitosan and alginate hydrogels. MTT assay results showed that ADSCs proliferated better on the composite hydrogels than on the chitosan and alginate hydrogels. Light microscope observation and cell cytoskeleton staining showed that hydrogel strength had obvious effects on cell growth and adhesion. The ADSCs seeded on chitosan and alginate hydrogels plunged into the hydrogels and could not stretch out due to the low strength of the hydrogel, whereas cells seeded on composite hydrogels with higher elastic modulus, could spread out, and grew in size.

Conclusion

The gelatin-polysaccharide composite hydrogels could serve as attractive biomaterials for tissue engineering due to their easy preparation and favorable biophysical properties.

Introduction

Biomaterials for tissue engineering have been widely studied, and play a pivotal role in providing platforms that facilitate cell adhesion, growth, and proliferation. However, one of the major challenges in designing functional scaffolds is to modify their properties to mimic the extracellular matrix (ECM) in the native tissues. The composition of ECM includes structural proteins, adhesion proteins, anti-adhesion proteins, and proteoglycans. The natural biomaterials used for the engineering of tissue constructs show many obvious benefits for mimicking ECM, including collagen, gelatin (GA), hyaluronic acid, laminin, chitosan, and alginate (Schwach & Passier, 2019). Polysaccharides can increase the stability of scaffolds, whereas proteins can enhance the biological properties. Therefore, mixing protein components (such as GA) with polysaccharide components (such as chitosan and alginate) to form composite materials mimicking natural ECM has become an important strategy for tissue engineering applications (Afewerki et al., 2019).

GA as a degradation product of collagen has good biocompatibility, high hydration degree, and low market cost. Therefore, GA has become a well-known biological material. However, natural GA hydrogel has low mechanical stability, thereby severely limiting its application in tissue engineering. Several methods have been used to overcome its defects, including physical mixing, chemical crosslinking, and enzymatic crosslinking. Chemical crosslinking agents, such as glutaraldehyde and methacrylic anhydride, were used to crosslink the composite materials of GA and polysaccharides to obtain better mechanical properties (Majidi et al., 2018; Miranda et al., 2011; Rosellini et al., 2009). However, incomplete removal of chemical crosslinking agents may lead to cytotoxicity (Li, Liu & Liu, 2009). Therefore, in our previous study, microbial transglutaminase (mTG) was used in place of chemical crosslinking agents to crosslink the composite materials, thereby resolving the problem of cytotoxic side effects (Yang et al., 2016).

Chitosan is a polysaccharide composed of glucosamine and N-acetyl-glucosamine. It is obtained by removing some acetyl groups from chitin, and it also an analog of glycosaminoglycan, which is a component of ECM. Therefore, chitosan exhibits many interesting biological properties, including biocompatibility, hydrophilicity, antibacterial, and antithrombotic effects. Chenite et al. (2000) first reported that nearly neutral chitosan/ β-GP aqueous solutions could gel quickly when heated. Thermosensitive chitosan hydrogel shows great potential in tissue engineering (Zhou et al., 2015). However, some authors reported that the thermosensitive chitosan hydrogel has insufficient mechanical strength when used as a tissue engineering material (Sacco et al., 2018; Supper et al., 2014; Zhang et al., 2019). Huang et al. (2014) constructed composite scaffolds comprising a chitosan hydrogel system and demineralized bone matrix, which exhibited an increased mechanical strength; the bone marrow stem cell (BMSC) retention of the hybrid scaffolds was more efficient and uniform than that of the other materials. Song et al. (2010) reported that ADSCs within the chitosan/ β-GP/collagen hydrogels displayed a typical adherent cell morphology and good proliferation with very high cellular viability after 7 days of culture. These experimental results indicated the possibility of overcoming the defects of chitosan gel by mixing with other materials, and composite scaffolds based on chitosan may be promising candidates for tissue engineering.

Alginate is a natural polysaccharide isolated from brown algae and bacteria. It has good biocompatibility and non-antigenicity; it is also antithrombotic and biodegradable. Besides, alginate has the advantages of having abundant sources and being low cost. Among the strategies used for the obtention of alginate hydrogel, the most widespread is ionic crosslinking (Sun & Tan, 2013). In the presence of multivalent cations, crosslinking is instantaneous and almost temperature-independent and allows solution/gel transformation under relatively mild conditions (Cattelan et al., 2020). Alginate hydrogel has been widely used in tissue engineering and has been approved for phase II clinical trials in the treatment of MI (Lee & Mooney, 2012). However, alginate hydrogel has natural poor cell adhesion and poor in vivo degradation performance (Bedian et al., 2017; Tønnesen & Karlsen, 2002). To solve these problems, mixing alginate with natural proteins, such as collagen, fibronectin, and GA, has been proposed (Hernández-González, Téllez-Jurado & Rodríguez-Lorenzo, 2020). Complexes of alginate and GA are interesting to use in tissue engineering, because they can provide suitable biological cues for hosting a variety of cells, including C2C12 myoblasts (Rosellini et al., 2018), HL1 cardiac muscle cell (Xu et al., 2009), neonatal rat ventricular myocytes (NRVMs) (Möller et al., 2011), and fetal rat myoblast H9C2 (Saberianpour et al., 2019).

Composite methods of GA and polysaccharides have been proposed, and the materials obtained through such methods support cell adhesion, proliferation, and differentiation (Miranda et al., 2011; Rosellini et al., 2009; Rosellini et al., 2018). Generally, strategies for preparing composite hydrogels include physical mixing (Liu et al., 2013), crosslinking (Rosellini et al., 2019), in situ synthesis (Wang et al., 2009), bio-conjugation (Ahadian et al., 2015), and others. In our composite hydrogels, enzymatic modification of proteins mediated by mTG is applied and aims to improve the properties of target products. These enzymatic reactions occur under mild reaction conditions and do not produce toxic products (Fatima & Khare, 2018). Da Silva et al. (2014) proved the feasibility of using mTG as a crosslinking agent for chitosan and GA hydrogel. Moreover, the microstructure of alginate GA-hydrogel microspheres was confirmed to be affected by GA content and mTG concentration (Pilipenko et al., 2019).

Adipose-derived stromal stem cells (ADSCs) are adult stem cells with abundant cell sources and provide a potential source of stem cells for tissue engineering research and clinical application (Suzuki et al., 2015). In this study, the preparation of chitosan/mTG-crosslinked GA (C-mTG/GA) and alginate/mTG-crosslinked GA (A-mTG/GA) was proposed, which are two kinds of composite hydrogels. Swelling, enzymatic degradation, and mechanical tests were performed. Viability and proliferation tests of ADSCs seeded on the composite hydrogels were conducted to determine the applicability of the hydrogels in tissue engineering. Finally, the cytoskeleton features of the ADSCs’ distribution on hydrogels were evaluated (Fig. 1B).

Figure 1 Schematic elucidating the preparation and characterization of the five hydrogels.

(A) Schematic showing the five hydrogels fabrication process. (1) Preparation of chitosan hydrogel. (2) Preparation of alginate hydrogel. (3) Preparation of mTG/GA hydrogel. (4) Preparation of two composite hydrogels. (B) Characterization and biological assessments of the five hydrogels. (C–G) The appearance of the five tested hydrogels. Scale bar = 1 cm.

Materials & Methods

Materials

Microbe transglutaminase (mTG, Bomei, China; enzyme activity, >100 U per gram), high-glucose Dulbecco’s modified Eagle’s medium (DMEM, Hyclone, UT, USA), fetal bovine serum (FBS, Gibco, NY, USA), and penicillin/streptomycin (P/S; Hyclone, UT, USA) were used. Trypsin (250 NFU/mg), ethylene diamine tetraacetic acid (EDTA), and collagenase type I (>125 NFU/mg) were purchased from Invitrogen (CA, USA). GA (type A, 300 Bloom), beta-sodium glycerophosphate (β-GP), calcein-AM, propidium iodide (PI), thiazolyl blue tetrazolium bromide (MTT), dimethyl sulfoxide (DMSO), aqueous formaldehyde, Triton X-100, 4,6-diamidino-2-phenylindole (DAPI), and fluorescein isothiocyanate (FITC)-phalloidin were purchased from Sigma (MO, USA). Chitosan (molecular weight: 100–300 kDa, degree of deacetylation: ≥85%), sodium alginate (Mw = 220,000 and M/G ratio = 0.38), calcium chloride (CaCl2), sodium chloride (NaCl), and acetic acid were purchased from Kelong (Chengdu, China). Analytical- or chemical-grade reagents were used.

Preparation of hydrogels

Preparation of chitosan hydrogel

Preparation of chitosan hydrogel was performed according to the method described elsewhere (Chenite et al., 2000) with minor modifications. Chitosan was placed on a piece of weighing paper, spread out, and exposed to ultraviolet (UV) light for 45 min, during which it was slightly turned over 3 times. The UV-disinfected chitosan particles were transferred to a sterile clean bench and dissolved in 0.1 M acetic acid. After stirring with a glass rod, it was placed at 4 °C for 24 h and centrifuged at 8,000 rpm for 10 min. The supernatant was collected to obtain the final chitosan solution concentration of 2% (w/v, weight volume ratio). This solution was stored at 4 °C. β-GP was dissolved in deionized water to obtain 50% (wt, weight ratio) solution, sterilized and passed through a 0.22 µm filter, and stored at 4 °C. During the preparation of the chitosan hydrogel, the β-GP solution was slowly dripped into the chitosan solution on the ice bag. The ratio of chitosan solution to β-GP solution was 5:1 (Xia et al., 2010), and the pH value was adjusted to 7.4. The mixed solution at 200 µL was added into a 24-well tissue culture plate (TCP) and placed into each well. The solution was incubated at 37 °C for 10 min to obtain chitosan hydrogel (Fig. 1A).

Preparation of alginate hydrogel

After disinfection by UV (as mentioned above), sodium alginate was dissolved in sterilized phosphate buffered saline (PBS) solution to form 1% (w/v) solution. CaCl2 and NaCl were mixed with deionized water to form solutions at final concentrations of 100 and 150 mM, respectively. After high-pressure steam sterilization, CaCl2/NaCl solution was stored at 4 °C. To prepare the alginate hydrogel, the sodium alginate solution was placed into a 24-well TCP (200 µL placed in each well). Then, 1 mL of CaCl2/NaCl solution was slowly dropped into each well. After soaking for 10 min at room temperature, the excess liquid was removed, and sodium alginate hydrogel was obtained (Fig. 1A).

Preparation of mTG/GA hydrogel

mTG/GA hydrogel was prepared using a protocol described in our previous publication (Long et al., 2017; Yang et al., 2016). GA was dissolved in PBS at 50 °C to obtain a solution, which was sterilized and passed through a 0.22 µm filter. mTG was dissolved in PBS to prepare 10% (wt) solution, which was sterilized and passed through a 0.22 µm filter. mTG/GA solution was prepared by adding mTG into the GA solution at 10 U/g⋅pro (enzymatic activity unit per gram of protein). Then, the mixed solution with a final GA concentration of 6% (w/v) was added into a 24-well TCP; 200 µL of the solution was placed into each well. The sample was incubated at 37 °C for 30 min for gelling (Fig. 1A).

Preparation of C-mTG/GA and A-mTG/GA hydrogels

mTG/GA, chitosan, and sodium alginate solutions were prepared as mentioned above. The 7.5% (w/v) GA with 10 U/g⋅pro mTG and 2% (w/v) chitosan (or 1% (w/v) sodium alginate) solutions were mixed in a volume ratio of 4:1. The composite process was conducted on a hot platform (DB-H, Xinbao, China) at 37 °C. The mixtures with a final GA concentration of 6% (w/v) were added into 24-well TCPs; 200 µL of the mixtures was placed into each well. The solutions were incubated at 37 °C for 20min to obtain C-mTG/GA and A-mTG/GA hydrogels (Fig. 1A).

Characterization of hydrogels

Gelation time

Ungelled mTG/GA mixture, β-GP/chitosan, C-mTG/GA, and A-mTG/GA solutions (1 mL) were added into 2 mL microtubes and incubated at 37 °C for gel. Sodium alginate solution at 1 mL was added into a 2 mL microtube, and CaCl2/NaCl solution was dropped for gelling. The onset of gelling was recorded as the gelation time, which was detected through the vial inverting method.

Hydrogel degradation test

In vitro enzymatic degradation property.

The in vitro enzymatic degradation property of hydrogels was evaluated by exposing them to enzymes to assess degradation rate. Collagenase has been used previously as a mimic for some of the protease secreted by cells (Mazzeo et al., 2019), and trypsin is often used in cell isolation and culture. The material degradation process of these proteases must be evaluated to provide a basis for cellular inoculation and digestion on hydrogels. The pre-weighed hydrogels (w0) were then immersed in 0.1% collagenase type I and 0.25% trypsin/0.01% EDTA for 12 h. At each time point (0.25, 0.5, 1, 2, 3, 4, 5, 6, 7, 8, 9, 10, 11, and 12 h), the liquid was removed completely. The hydrogels were weighed again (wt). The degree of degradation (D) was calculated as follows: D%=w0−wt∕w0×100.

Three repeated measurements were performed for each type of hydrogel.

Hydrolytic and cellular degradation.

The hydrolytic and cellular degradation of hydrogels were performed as described by our previous study (Yang et al., 2016). In brief, hydrogels were prepared into 35 mm culture dishes with ∼1 mL for each dish. To exclude the influence of swelling behavior of hydrogels, 2 mL of PBS was added into each dish and PBS was removed completely after 12 h; the hydrogels were weighed (w0). The hydrogels were incubated in PBS at 37 °C with 5% CO2 for two weeks. PBS was changed every 2 days. At different time points (0, 2, 4, 6, 8, 10, 12, and 14 d), three samples of each kind of hydrogel were weighed (wt). The degree of degradation was calculated as above. Cell-mediated degradation was measured by seeding 1.0 × 106 ADSCs (see ‘Primary culture of ADSCs’) on each hydrogel scaffold. The rest of the steps were the same as above, except that the PBS was changed to cell culture medium (high-glucose DMEM, 15% FBS, and 1% P/S).

Swelling capacity

Each type of hydrogel was prepared and weighed (w1). The hydrogels were immersed into PBS for 12 h at 37 °C, and excess PBS was blotted out with filter paper. The swollen hydrogels were obtained and re-weighed (w2). Swelling ratio (S) was calculated as follows: S%=w2−w1∕w1×100.

Three repeated measurements were performed for each type of hydrogel.

Mechanical property

The mechanical property of the hydrogels was evaluated using a mechanical testing apparatus (HPB, Handpi, China). The effects of sample cutting were considered. Only the samples with the same dimensions were selected as test specimens. For this purpose, cylinder-shaped samples were cut to achieve a diameter of 15 mm and a thickness of 6 mm. The sample was fixed on a hot platform and examined at 37 °C. The detecting probe was a stainless steel cylinder (12.5 mm in diameter) with a flat front attached to the mechanical testing machine. The hydrogels were compressed at a constant deformation rate of 1.0 mm/s. Meanwhile, the value of loading force was recorded automatically by using a mechanical testing software (Handpi, China). The slopes of compressive stress–strain curves at 0% to 50% deformation were used to calculate the compressive modulus. The reported values are the mean of six specimens.

Primary culture of ADSCs

The study protocol was approved by the Institutional Animal Care and Use Committee of Sichuan University (approval number: KS2019006). ADSCs were isolated as described previously (Yang et al., 2016). Subcutaneous adipose tissue was obtained from a Sprague–Dawley rat (weighed 100 g, either male or female) and finely minced. The minced tissue was placed in a digestion solution containing 0.1% collagenase type I and subjected to continuous agitation at 37 °C for 45 min. The cell suspension was filtered and centrifuged at 2000 rpm for 5 min. Cellular precipitation was resuspended with the cell culture medium and cultured in 25 mm2 cell flasks. The cells were cultured at 37 °C in a 5% CO2 incubator, and the medium was changed twice a week. Cultures were passaged every 5 days. The cells were observed daily under an inverted phase-contrast microscope (CKX41, Olympus, JAPAN). The cells were detached with 0.25% trypsin/0.01% EDTA and re-plated for cell passage. The third-passage ADSCs were used for the subsequent experiments.

Cell viability

2D culture.

Different types of hydrogels were prepared as described above. After washing with PBS, the hydrogels were ready for cell culture. ADSCs were seeded on the surface of the hydrogels at 1.0 × 104 cells per well. ADSCs at same number were seeded on non-hydrogel TCPs as controls.

3D culture.

ADSCs were prepared as described above and cell density was adjusted to 5.0 × 106 cells/ml. The cell/hydrogel mixtures were obtained by mixing cell suspensions and different hydrogel solutions at 1:9 volume ratio and the mixtures were pipetted into the wells of a 24-well culture plate at 100 µL per well (∼5.0 × 104 cells per well). The cell/hydrogel constructs were incubated at 37 °C for 2 h and then supplemented with cell culture medium.

On day 5 of ADSC culture, the cell-cultured samples were washed thrice with PBS and incubated in 250 µL PBS containing 2 µM calcein-AM and 2 µM PI at 37 °C for 30 min. After re-washing with PBS, the cells were observed by using an inverted fluorescent microscope (XDS30; Sunny Instruments, China) equipped with a color digital camera (MD50; Mingmei, China). The viability of ADSCs was determined by staining with calcein-AM and with PI to label the live and dead cells, respectively.

Cell proliferation studies

ADSCs were seeded on the hydrogels and TCPs as described above at 1 × 104 cells per well. Cell proliferation was determined by MTT assay on days 0, 2, 4, and 6 of cell culture. At each time interval, the cell-cultured samples (three replicates) were rinsed thrice with PBS and treated with 800 µL of high-glucose DMEM containing 80 µL of MTT solution (5 mg/mL in PBS) at 37 °C for 4 h. The supernatant was removed after incubation, and the formazan crystals in the cells were dissolved in 400 µL of DMSO. Then, the absorbance of 100 µL of supernatant transferred to a new 96-well TCP was measured at 490 nm with a reference wavelength of 630 nm by using a microplate reader (Biotek ELx800, USA). Background absorbance from the control wells, which contained the culture medium but without cells, was subtracted.

Cytoskeleton staining

Cells were seeded on the hydrogels and TCPs at 1 × 105 cells per well (6-well TCPs) and cultured for 5 days. Cells were fixed with 4% (w/v) aqueous formaldehyde solution for 15 min at room temperature, and permeabilized with 0.1% (v/v) Triton X-100 solution in PBS for 10 min. Afterward, they were stained with 5 µg/mL FITC- for 30 min followed by 1 µg/mL DAPI for 10 min. After incubation, fluorescent images were acquired using the inverted fluorescent microscope equipped with a digital camera.

Statistical analysis

Statistical analyses were performed using SPSS software (SPSS Inc.). Each experiment was repeated ≥3 times. Data are presented as mean with standard deviation. Statistical significance between two groups was determined by an independent sample Student’s t-test. The level of statistical significance was P < 0.05.

Results

Gelation time and appearance of hydrogels

The chitosan samples at 1 mL of 2% (w/v) gelatinized after the addition of β-GP solution within 10 min at 37 °C, and the obtained hydrogel was opaque and light yellow. The GA samples at 1 mL of 6% (w/v) needed approximately 30 min to achieve gelling at 37 °C. The mTG/GA hydrogel was colorless and transparent. This result was consistent with that obtained in our previous study (Yang et al., 2016), in which we reported that the 6% solution of GA mixed with mTG took 25 ± 0.55 min to gel. Alginate hydrogel samples at 1 mL of 1% (w/v) were obtained by Ca2+-crosslinking at room temperature within 10 min. When the calcium solution was dropped into the alginate solution, the edge started to shrink immediately and form wrinkles. After removing the additional liquid, we obtained a translucent white alginate hydrogel with uneven thickness. Samples of the two composite solutions (1 mL) prepared as mentioned in ‘Preparation of C-mTG/GA and A-mTG/GA hydrogels’ gelatinized within 20 min at 37 °C. The C-mTG/GA hydrogel was translucent yellow. The A-mTG/GA hydrogel was colorless and transparent (Figs. 1C–1G).

Swelling capacity

Swelling capacity reflects the hydrophilic character and water retention capacity of hydrogels, and is also an important factor for predicting nutrient transfer within hydrogels (Gu et al., 2020). After being immersed in PBS for 24 h, mTG/GA, C-mTG/GA, and A-mTG/GA hydrogels were slightly expanded, whereas the volumes of chitosan and alginates hydrogels almost remained unchanged. Swelling ability is an important index for hydrogels that are used in tissue engineering. We evaluated the swelling rate of the five hydrogels (Table 1). Chitosan and alginate hydrogels showed maximum swelling rates of 1.55 ± 0.89% and 4.68 ± 0.87%, respectively, in PBS. Both swelling rates were significantly lower than that of the other three hydrogels. The swelling rate of mTG/GA hydrogel was 21.33 ± 3.56%. Although with the same concentration of GA, the swelling rate of C-mTG/GA hydrogel decreased to 14.29 ± 2.28% (P < 0.05). The swelling rate of A-mTG/GA hydrogel (20.72 ± 0.84%) did not differ significantly from that of mTG/GA hydrogel (21.33 ± 1.78%, P = 0.62 >0.05).

Table 1 Gelation time, swelling rate and compression modulus of the five tested hydrogels.

Samples	Chitosan hydrogels	Alginate hydrogels	mTG/GA hydrogels	C-mTG/GA hydrogels	A-mTG/GA hydrogels	
Gelation time (min)	<10	<10	∼30	<20	<20	
Swelling rate (%)	1.55 ± 0.89*,**	4.68 ± 0.87*,**	21.33 ± 1.78**	14.29 ± 2.28*,**	20.72 ± 0.84*	
Compression modulus (kPa)	3.48 ± 0.45***	7.06 ± 1.22***	17.42 ± 1.34***	14.29 ± 2.64***	19.79 ± 1.22***	
Notes.

*,**,*** P < 0.05

Mechanical property

As cardiac tissue engineering substrates, the hydrogels need to have proper mechanical properties. We used a mechanical testing machine to evaluate the compression modulus (Table 1). Chitosan and alginate hydrogels exhibited low compression moduli (3.48 ± 0.9 and 7.06 ± 2.44 kPa). Compared with mTG/GA hydrogel (17.42 ± 2.68 kPa), the strength of the composite hydrogel showed a decreasing trend after mixing with chitosan (14.29 ± 5.28 kPa, P < 0.05), whereas the strength of the composite hydrogel showed an increased trend after mixing with sodium alginate (19.79 ± 2.44 kPa, P < 0.05). These results may be related to the formation of intermolecular interactions (electrostatic interaction and hydrogen bonding) between the polypeptide chains of GA and the macromolecules of polysaccharides. The additional junctions in the complex gel network result in changes in elasticity compared with those of native GA (Derkach et al., 2020).

Hydrogel degradation test

Cells can produce different proteolytic enzymes via autocrine and/or paracrine, which may lead to the degradation of hydrogel. Therefore, the in vitro enzymatic degradation of hydrogels should be evaluated. Degradation tests in two parallel groups were conducted, namely, collagenase and trypsin degradation groups. The curves of enzymatic degradation are shown in Fig. 2. In the collagenase degradation tests, the C-mTG/GA hydrogel showed the fastest degradation rate, in which more than 30% of the original weight was lost in 15 min, and only 0.20 ± 0.36% remained after 5 h. During degradation by collagenases, mTG/GA and A-mTG/GA hydrogels also showed obvious degradation. Complete degradation of mTG/GA hydrogels occurred within 6 h, and complete degradation of A-mTG/GA hydrogels required 7 h. However, the enzymolysis rates of chitosan and alginate hydrogels were markedly slower than those of the other three hydrogels. For alginate hydrogel, only 13.76 ± 3.57% of the hydrogel was lost after 12 h. For chitosan hydrogel, approximately 96% remaining after digestion for 12 h.

Figure 2 In vitro enzymatic degradation property of the five hydrogels.

(A) The curves of 0.1% collagenase degradation. (B) The curves of 0.25% trypsin/0.01% EDTA degradation. (C) The curves of hydrolysis degradation. (D) The curves of cellular degradation.

Based on the trypsin degradation tests, mTG/GA and C-mTG/GA hydrogels almost completely dissolved after 2 h of enzymatic degradation. A-mTG/GA hydrogels showed slower degradation rate and were degraded completely by trypsin within 8 h. Alginate hydrogels showed nearly half of the mass loss after 12 h. However, chitosan hydrogel did not show degradation after 1 h, and its final mass after 12 h remained at approximately 90%.

All hydrogels exhibited high capacity for hydrolytic resistance. After two weeks of immersion, all of the five kinds of hydrogels retained more than 95% of their original mass. In the test of cell-containing hydrogels, we found that all hydrogels degraded more rapidly than cell-free hydrogels. For two weeks, 25.1 ± 3.48% of mTG/GA hydrogel mass was lost. Moreover, the alginate hydrogel also showed significant degradation. At the second week, 12.2 ± 2.99% of the hydrogel mass was lost. However, the mTG/GA, C-mTG/GA, and A-mTG/GA hydrogels did not show severe degradation. For C-mTG/GA, and A-mTG/GA hydrogels, approximately 5% of gel mass was lost after two weeks of incubation; meanwhile for chitosan hydrogel, mass loss was 1.2 ± 0.58%. These results may suggest that the incorporation of polysaccharides can help strengthen cellular degradation resistant capacity of GA-containing hydrogels.

Cell morphological observation

To observe the growth and adhesion of ADSCs on hydrogel surface and verify the biocompatibility of the hydrogels, cell images were recorded by using the inverted phase contrast microscope. Figure 3 shows the cell morphology of 2D cultures on the five hydrogels for 2 h, 1 day, and 3 days. After inoculation for 2 h, most of the ADSCs on the TCP were already attached and extended to the plate. Cells on the mTG/GA, C-mTG/GA, and A-mTG/GA hydrogels showed some pseudopods and were stellate or irregular in shape. Some rounded cells still not attached to the surface of chitosan and alginate hydrogels were present. After inoculation for 1 day, most cells showed spreading activity, and cell proliferation was observed on the surface of C-mTG/GA and A-mTG/GA hydrogels. The elongation of ADSCs was also observed on the surface of mTG/GA hydrogel. Nevertheless, the cells on the surface of the chitosan and alginate hydrogels remained round in shape. Some of the cells on the surface of these hydrogels were surrounded by a ring shadow, indicating that cells were moving in and out within a relatively small space. These cells were trying to stretch out their pseudo feet. However, because no cell adhesion sites were present, these cells were unable to adhere to the surrounding hydrogel tightly, and hardly any stretching was maintained. Therefore, when these shadows appear, the range of cell activity can be inferred indirectly. The images at day 3 clearly showed the cells reached confluence and covered the surface of C-mTG/GA and A-mTG/GA hydrogels. The cells on the surface of chitosan hydrogel grew inward because of the inadequate strength for cell spreading. A few of the polygonal ADSCs were recorded on the surface of the alginate hydrogel.

Cell viability

The viability of ADSCs on the hydrogels was determined using live/dead staining assay by imaging live and dead cells under a fluorescent microscope. Cells that lost membrane integrity and were no longer viable were stained red (dead cells), whereas the viable cells were stained green (live cells) (Fig. 4). The five hydrogels show good biocompatibility and were suitable for cell 2D culture. Although the proportion of dead cells was small in 2D cultures, cells seeded on chitosan and alginate hydrogel displayed negative growth (10–20 live cells per visual field at 10 × magnification) over the 5 days and remained circular in shape. The cells on mTG/GA, C-mTG/GA, and A-mTG/GA hydrogels grew in size and clustered together. The numbers of live ADSCs cultured on mTG/GA and A-mTG/GA hydrogels in each visual field were more than 100 cells, and the numbers of dead cells in same visual fields were less than 10 cells. The average quantity of live cells on C-mTG/GA hydrogel was slightly less (∼75 live cells per visual field) than on the other hydrogels. Cell shape progressively assumed prickly or rhombohedral patterns. The cell morphology indicated that the cell development may progress to achieve a 3D shape instead of a flat 2D shape. The ADSCs on the C-mTG/GA hydrogel appeared to stretch out to different directions in a 3D hydrogel space (the circle of Fig. 4D). The ADSCs developed on TCPs showed higher quantity and a spreading shape.

Figure 3 Cell growth at 2 h, 1 day, and 3 days were observed by using an inverted phase contrast microscope.

(A–C) Chitosan hydrogels, (D–F) alginate hydrogels, (G–I) mTG/GA hydrogels, (J–L) C-mTG/GA hydrogels, (M–O) A-mTG/GA hydrogels, and (P–R) TCPs. Scale bar = 200 µm.

Figure 4 Live/dead cell staining results after 5 days of culture.

The cytoplast of live cells emitted green fluorescence when stained with calcein-AM. The nuclei of dead cells emitted red fluorescence when stained with PI. (A) chitosan hydrogels, (B) alginate hydrogels, (C) mTG/GA hydrogels, (D) C-mTG/GA hydrogels, (E) A-mTG/GA hydrogels, and (F) TCPs. Scale bar = 200 µm.

Live/dead staining of ADSCs embedded in the five hydrogels, shown in Fig. 5, revealed that >90% of cells survived the 3D culture. At day 5 in culture, it was evident that GA-containing hydrogels promoted spreading of embedded cells and resulted in more surviving cells compared to chitosan and alginate hydrogels. According to the images recorded by the inverted fluorescent microscope, ADSCs remained a rounded morphology in chitosan and alginate hydrogels after 5 days of culture. While in mTG/GA, C-mTG/GA, and A-mTG/GA hydrogels, the cells assumed barbed-like morphology with numerous cell protrusions stretching out in different directions, which represented the three hydrogels supporting the cell survival and adhesion.

Figure 5 Observing cell viability in 3D cultures at 5 days.

(A–C) Chitosan hydrogels, (D–F) alginate hydrogels, (G–I) mTG/GA hydrogels, (J–L) C-mTG/GA hydrogels, (M–O) A-mTG/GA hydrogels. Scale bar = 200 µm.

MTT assay results

Significant difference in cell growth behavior was observed from day 0 to day 6 (Fig. 6). Although the same number of cells were seeded on day 0, the number of cells cultured on each hydrogel was less than that of the TCP control group at different time intervals, and the difference was statistically significant (P < 0.05). There was an adaptation period, during which the cells grew on the materials. Some cells were detached from hydrogels. As a result, the number of cells on the hydrogels was certainly less than that of the control group. However, if the duration of measurement was longer, then the number of cells on the materials may increase. In our previous experiments (Yang et al., 2016), the number of cells growing on the materials could exceed the number of cells on TCPs with increasing measurement time. The purpose of this study was to determine the effects of different materials on cell compatibility and proliferation, but not to compare the difference of cell growth behavior between hydrogels and the control group. Therefore, long-term MTT assay (>2 weeks) was not performed. Compared with ADSCs on alginate and chitosan hydrogels, those on mTG/GA and the two composite hydrogels showed more cells and a faster proliferation rate. A decrease in cell proliferation was observed from samples on the chitosan and alginate hydrogels on day 4, and such a decrease may have been due to the poor adhesion of the cells on these two materials. Some cells were lost when the culture medium was changed routinely on day 3.

Figure 6 The proliferation of the ADSC cultured on the hydrogels and TCPs after 0, 2, 4, and 6 days of culture, as determined by MTT assay.

Cytoskeleton staining

Cytoskeleton is a network system of protein fibers in eukaryotic cells. In a narrow sense, the cytoskeleton is composed of microtubules, microfilaments, and intermediate fibers. Microfilaments are spiral fibers composed of filamentous actin (F-actin). When the adherent cells spread out and became larger on the culture surface, the expression of actin increased, which led to the formation of the actin network. FITC-phalloidin is a kind of microfilament depolymerization inhibitor that has a strong affinity with actin filaments and only binds to F-actin. Therefore, only the F-actin of the ADSCs seeded on the hydrogels with excellent properties in cell adhesion and growth can be examined with FITC-phalloidin.

Figure 7 shows the spreading and morphology of ADSCs grown on hydrogels and TCPs. ADSCs cultured on chitosan hydrogel for 5 days were still round and had no obvious cytoskeletal structure. The chitosan hydrogel showed poor properties for cell adhesion and growth, and the myofilament structure did not form in the cells. Few ADSCs on alginate hydrogel were polygonal, and few actin fibers were observed on the edge of the cells. ADSCs on C-mTG/GA hydrogel showed a radial arrangement of microfilament cytoskeleton. The staining images of ADSCs on A-mTG/GA and mTG/GA hydrogels were similar to those of the cells on TCPs. Most cells spread on the surface of the material, and the cytoskeleton network was orderly arranged.

Figure 7 Cytoskeleton staining.

The F-actin emitted green fluorescence when stained with FITC-phalloidin. The nuclei emitted blue fluorescence when stained with DAPI. (A) chitosan hydrogels, (B) alginate hydrogels, (C) mTG/GA hydrogels, (D) C-mTG/GA hydrogels, (E) A-mTG/GA hydrogels, and (F) TCP. Scale bar = 100 µm.

Discussion

To better mimic the physiological, biochemical, and physical cues of native tissues, the hybrid materials have extensively been explored. At a minimum, the preferred biomaterial for tissue engineering needs to meet the following essential criteria (Yue et al., 2020; Zhang et al., 2019): (1) biodegradability, (2) proper elastic modulus, and (3) good biocompatibility.

In the field of regenerative medicine, the scaffold material usually needs to be biodegradable (Wang et al., 2019). However, some exceptions exist, such as bone, articular cartilage, or cornea tissue engineering, which require stability of the implanted material (Rastogi & Kandasubramanian, 2019). The blending of polymers may affect the degradation behavior. Therefore, degradation test of the scaffolds was conducted in vitro. Under the action of collagenase, the three hydrogels of mTG/GA, C-mTG/GA, and A-mTG/GA showed similar enzymatic degradation rates. Upon addition of trypsin, both mTG/GA and C-mTG/GA completely degraded in a short time. These results indicated that enzymatic cross-linking can provide stability of the scaffold but does not hinder the degradation of protein components. However, for A-mTG/GA hydrogels, the speed of enzymatic degradation by trypsin is slower than that of mTG/GA and C-mTG/GA hydrogels, which may be due to the fact that sodium alginate, as a natural anionic polymer, prevents the trypsin activity from entering GA through electrostatic action (Lv et al., 2014). Ruvinov et al. also observed that alginate-sulfate hydrogel protected the protein from the hydrolysis of trypsin (Ruvinov, Leor & Cohen, 2010). GA and chitosan are biodegradable, whereas alginate shows high stability in vivo (Bedian et al., 2017). In our work, chitosan and alginate hydrogels were more stable to degradation of the two proteases, because the main components of chitosan and alginate gels are polysaccharides. However, alginate hydrogel lost almost half of its mass under the action of trypsin after 12 h. This phenomenon might be related to the addition of 0.01% EDTA to trypsin. Dimerization of alginate chains is induced by calcium, and as a result, gel networks are formed. Depending on the amount of calcium present in the system, these inter-chain associations can be either temporary or permanent (George & Abraham, 2006). EDTA can chelate Ca2+ (Hafer et al., 2020); the content of calcium in the A-mTG/GA system is reduced, resulting in a thixotropic solution with high viscosity (George & Abraham, 2006), thereby finally showing the decline of solid mass.

The mechanical property of hydrogel governs final tissue engineering usage, e.g., less stiffness and softer hydrogel can be used to soft tissues of the brain, and high stiffness and harder hydrogel may be effective for hard tissues of bones, thereby prompting us to characterize the mechanical properties of the five types of hydrogels in this study. Our results showed that the mTG/GA, C-mTG/GA, and A-mTG/GA hydrogels had higher elastic moduli than that of chitosan and sodium alginate hydrogel. The addition of polysaccharides particles influenced the elastic modulus, while the effect differs depending on the type of polysaccharides used. After adding chitosan to GA, the structural heterogeneities in the composite’s network cause descending of the hydrogel’s mechanical property, but after adding sodium alginate to GA, the raising of the elastic modulus indicate the increase of physical crosslinking density (Lewandowska-Łańcucka et al., 2017; Li et al., 2018). The stiffness of substrate also aids in cell functioning, i.e., in growing and proliferating (Caliari & Burdick, 2016). ADSCs grown on GA and on the two kinds of mixed hydrogels showed better adhesion and elongation, whereas ADSCs grown on alginate hydrogel showed relatively later adhesion. ADSCs cultured on chitosan hydrogel cannot adhere and grow due to low elastic modulus. Cytoskeleton staining also confirmed this conclusion. The cell morphology of ADSCs on alginate hydrogel was irregular without an obvious cluster-like branch. ADSCs cultured on chitosan hydrogel almost did not form filamentous actin, and the cell shape remained round.

Hydrogels are water-insoluble networks of crosslinked hydrophilic polymers that exhibit swelling capacity in aqueous environments. The ability of water retention by the material strongly depends on the microstructure (Yan et al., 2005). After mixing chitosan into GA, the mixed hydrogel showed a decrease in swelling rate, which may be attributed to the formation of the tight microstructure of chitosan and GA. The swelling rate of the mixed hydrogels of sodium alginate and GA did not change significantly compared with the mTG/GA hydrogel, indicating that alginate molecules had little effect on the microstructure. Nadezhda et al. also considered that the swelling capacity of the microspheres prepared by alginate and GA was mainly regulated by the content of GA (Lewandowska-Łańcucka et al., 2017). The higher swelling capacity of the hydrogels enhances cell proliferation and cell viability by facilitating transport of nutrients into the hydrogels (Annabi et al., 2011; Gu et al., 2020). That corresponds well with our findings.

When the scaffold is implanted into the body, material biocompatibility becomes a key issue. The scaffold material must not induce adverse reactions, sensitization, carcinogenicity, and irritation in cells, tissues, and systems of humans. The material is supposed to degrade itself. The degradation products need to be non-toxic and can be absorbed or metabolized by the human body. Therefore, whether biomaterials can be successfully applied in tissue engineering depends on the biocompatibility of materials and the toxicity of degradation products (Catalano et al., 2013). The biocompatibilities of chitosan, sodium alginate, and GA have been confirmed (Chatelet, Damour & Domard, 2001; Li et al., 1999; Sosnik, 2014); however, whether the blending of these three abovementioned polymers affect the biocompatibility of materials needs to be further studied. We found that the composite hydrogel had no cytotoxic effect on ADSCS according to staining results on live and dead cells. In addition to supporting the survival of ADSCs, growth curves based on MTT assay suggested that ADSCs can proliferate in mTG/GA, C-mTG/GA, and A-mTG/GA hydrogels. All these results indicated that the two composite hydrogels have good biocompatibility. The composite gels are obtained by physical mixing of GA and polysaccharides, thus they retain various cell-friendly active sites of GA, such as arginine-glycine-aspartic acid (Lee et al., 2003), which then provides cultured cells with a friendly environment for growth and proliferation.

Conclusions

To mimic the chemical composition of natural tissue, the combination of GA and chitosan or alginate was used to successfully fabricate composite hydrogels in this paper. Hydrogels used for cell culture must exhibit desirable characteristics, such as good swelling capacity, proper mechanical property, and biocompatibility. In this study, the physical properties of five hydrogels were assessed. The composite hydrogels showed different mechanical properties and swelling capacities that depended on different polysaccharide added. Most importantly, this study showed that mTG/GA, C-mTG/GA, and A-mTG/GA hydrogels have excellent biocompatibility and can support ADSC survival, adhesion, and proliferation. Therefore, we believe the biomimetic composite hydrogels of GA and polysaccharides could be suggested as promising materials to cell carriers in tissue engineering.

Supplemental Information

Supplemental Information 1 Compression modulus data

Click here for additional data file.

Supplemental Information 2 Enzymatic property data

Click here for additional data file.

Supplemental Information 3 Swelling rate data

Click here for additional data file.

Supplemental Information 4 Results of MTT assay

Click here for additional data file.

Supplemental Information 5 Supplementary experiment data

Raw data of supplementary experiment (Live/dead staining images of 3D cultures and results of degradation test).

Click here for additional data file.

Additional Information and Declarations

Competing Interests

Author Contributions

Animal Ethics

Data Availability

The authors declare there are no competing interests.

Jing Ye conceived and designed the experiments, performed the experiments, analyzed the data, prepared figures and/or tables, authored or reviewed drafts of the paper, and approved the final draft.

Gang Yang conceived and designed the experiments, performed the experiments, prepared figures and/or tables, authored or reviewed drafts of the paper, and approved the final draft.

Jing Zhang performed the experiments, prepared figures and/or tables, and approved the final draft.

Zhenghua Xiao conceived and designed the experiments, authored or reviewed drafts of the paper, and approved the final draft.

Ling He and Han Zhang analyzed the data, prepared figures and/or tables, and approved the final draft.

Qi Liu analyzed the data, authored or reviewed drafts of the paper, and approved the final draft.

The following information was supplied relating to ethical approvals (i.e., approving body and any reference numbers):

Institutional Animal Care and Use Committee of Sichuan University provided full approval for this research (approval number: KS2019006).

The following information was supplied regarding data availability:

Raw data are available in the Supplemental Files.

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
