# Peer review of "Preparation and characterization of gelatin-polysaccharide composite hydrogels for tissue engineering"

_PeerJ, doi:10.7717/peerj.11022_

## Round 0.1 · original submission · Major Revisions

Please address all the comments by the reviewers. In particular, a reviewer asks to go further into the "cardiac application". You have two options, either change the title to not consider cardiac applications or keep the title but fulfulling what is asked by the reviewers in this regard.
The manuscript is well written in English, few errors exist. Please review the manuscript again to find and fix these language issues. Moreover, as indicated by one of the reviewers, a figure will help understanding the preparation and differences of all hydrogels. My specific comments (to be addressed also) are:

1. Please add additional details of the employed sodium alginate

2. How was the 5:1 ratio established for chitosan to beta-GP?

3. How were the conditions selected for using beta-GP? Temperature, concentration, pH, time, etc; for all crosslinked gels.

4. Why were collagenase and trypsin selected for degradation studies?

5. Why only 2 h were analyzed in degradation studies? Is this representative for the intended application?

6. It is not clear what is the effect of the swelling capacity in the intended application. Is it desirable to have higher or smaller swelling capacity?

7. Figure 2 must be improved. The abbreviation for minutes is min.

8. Include SEM micrographs and FTIR analysis of the gels studied alone and when cells are growing.

9. I suppose that you included the degradation analysis as a model of what will happen in the intended application. In this regard, I suppose the stability of the gel must be improved. Run experiments with beta-GP at different conditions to show evidence that you can control this.

10. In the discussion section, as pointed by one of the reviewers, you must include your own discussion. Why are chitosan and alginate gels more stable to degradation than the composite counterparts? What is propitiating the differences in swelling capacity? Why there are differences in the moduli? Why is cell growth favored in the composite gels?

11. The conclusions are too general. A reference in your conclusions is odd. Conclude about what was studied in your work. "Good choices for tissue engineering" is unproven in this manuscript.

Reviewer 1 ·

Basic reporting

no comment

Experimental design

no comment

Validity of the findings

no comment

Additional comments

In this manuscript, the authors reported a series of gelatin-polysaccharide composite hydrogels and evaluated their potential in cardiac applications. The experiments are well-designed, and the results can support the conclusions of the authors. Overall, the quality of this manuscript is quite good. So, I think it can be accepted after some minor revisions.
1. A schematic diagram should be added to make the audience understand the idea of this work easier.
2. The quality of Figure 2 is rather poor. The authors should futher improve it.

Reviewer 2 ·

Basic reporting

Reported experiments were performed well. However, in the introduction section, authors are advised to write about the previous literature in the development of hydrogel for cardiac tissue regeneration. Title and introduction, and experiments performed were not correlated.
Molecular weight, mannuronic acid/ guluronic acid ratio and other characterization of alginate needs to be mentioned to materials section.

Experimental design

Authors are requested to perform experiments showing suitability of the development hydrogel for cardiac tissue regeneration. With only in vitro cell viability and attachment, it difficult understand how the developed hydrogel will be intended for cardiac tissue regeneration. Authors are required to perform invitro cardiogenic differentiation.
Additionally, cell migration/penetration inside hydrogel needs to performed, as the hydrogel is intended for tissue regeneration. Z-stacking in confocal microscopy could provide the same.
In vitro degradation studies show, composite hydrogel gets degraded in 2 hour. How's the cell culture experiments were performed? This experiment needs to discuss properly.
Gelation time for each hydrogel could be tabled for essay understanding for the readers.

Validity of the findings

Authors needs provide reason for selecting 4:1 ratio for gelatin and chitosan/alginate.

Reviewer 3 ·

Basic reporting

No comment

Experimental design

No comment

Validity of the findings

No comment

Additional comments

In this study, to detect the properties of composite hydrogels for cardiac tissue engineering in vitro, the authors developed five different hydrogels, including chitosan hydrogel, alginate hydrogel, microbial transglutaminase-crosslinked gelatin (mTG/GA) hydrogel, chitosan mTG-crosslinked GA (C-mTG/GA) hydrogel, and alginate mTG-crosslinked GA (A-mTG/GA) hydrogel. Then the authors assessed enzymatic degradation property in vitro, swelling capacity, and mechanical property of all hydrogels. In order to test the growth and adhesion of adipose-derived stromal stem cells (ADSCs) on the hydrogel surface, the cell morphology of ADSCs after inoculation in five hydrogels was observed by immunofluorescence staining. Calcein acetoxymethyl ester/propidium iodide (calcein-AM/PI) technology was used to measure cell viability. Besides, methyl tetrazolium (MTT) was used to test the cell proliferation. At last, the authors researched the cytoskeleton staining. This study suggests a unique approach to modernizing myocardial infarction (MI) through the composite hydrogel, and the authors also showed some significant results from the developed hydrogels on MI treatments. However, there are still a few issues to be clarified prior to the publication of this manuscript.
1. As shown in Table 1, the swelling capacities of mTG/GA, C-mTG/GA, and A-mTG/GA hydrogels are higher than those of chitosan and alginate hydrogels. In order to make this manuscript clearer and more significant, the authors should clarify if the swelling capacities would affect the security or efficacy of tissue engineering, and what is the exact relationship.
2. The authors chose the collagenase type I and trypsin to degrade the materials. However, as the materials used for cardiovascular applications, the authors didn’t explain why. As far as I know, the most common site of trypsin should be the small intestine.
3. Although the authors did the enzymatic degradation test, those hydrogels will be used for cardiovascular applications, should the authors further study the platelet agglutination. The reviewer thinks that If these materials are not entirely degraded before forming new blood vessels and cardiomyocytes, once platelets gather on these materials, they can cause a myocardial infarction.
4. Whether the specific physical and chemical properties of these composite hydrogels need further elaboration, such as transmission electron microscopy, Fourier transform infrared (FTIR) spectra, and so forth.
5. As for cell viability and cell proliferation test, the authors gave us the results that cells in the mTG/GA, C-mTG/GA, and A-mTG/GA hydrogels behaved better than those of chitosan and alginate hydrogels. However, the authors didn’t explain a clear reason or give your conjecture.
6. In the discussion section, the authors should use your own opinions rather than quoting others.
7. The layout of pages 24 and 24 needs further refinement.
8. The recently published review or research articles should be discussed in the revision, for example, Trends in Biotechnology 2020, 38 (6), 579-583; Biomacromolecules 2019, 20 (4), 1478-1492; Advanced Science 2018, 5 (5), 1700527; ACS Applied Materials & Interfaces 2019, 11 (9), 8725-8730.

Reviewer 4 ·

Basic reporting

The authors have developed composite hydrogels composed of gelatin and polysaccharide (chitosan or alginate) for tissue engineering applications. The enzymatic cross linking and simple method of development make the hydrogels attractive to take it further. The effort with which the paper is written is appreciated where the authors have considered to write each section in detail. However, some major revision has to be done to make this manuscript publishable. Please consider the points below:
1. The authors have used the term “we” in many places. I recommend rewriting the manuscript in passive form.
2. There is a need to do English language and grammar correction.

Experimental design

1. The title of the manuscript is misleading. The authors have titled “cardiac applications”, but the manuscript does not contain any experiments to prove the same. Differentiation experiments have to be done to claim the applicability of the material in cardiac tissue engineering.
2. The experiment of enzymatic degradation is done only for 2h and the results showed that most of the gel degraded in this short span. How will the authors use this gel for long term experiments? Authors should test for improved enzymatic crosslinking by altering conditions such as time, temperature etc so as to reduce fast degradation.

Validity of the findings

-No Comments-

---

## Round 0.2 · Minor Revisions

I would like to thank the authors for addressing most of the comments by the reviewers and myself. As asked by two reviewers, further degradability and survivability studies must be conducted, otherwise the potential of composite comes short. Please run these studies within your capabilities.

Reviewer 1 ·

Basic reporting

No comment

Experimental design

No comment

Validity of the findings

No comment

Additional comments

The authors have revised their manuscript carefully according to all reviewers' comments. I think this version is acceptable for publication.

Reviewer 2 ·

Basic reporting

No Comments

Experimental design

Authors had performed in vitro degradation studies under non-cellular enzymatic condition. Authors need to perform the degradation studies based on hydrolytic and Cell-mediated degradation for a long duration. This would help the readers to better understand the degradation properties of the developed hydrogel.

Validity of the findings

No comments

Reviewer 3 ·

Basic reporting

No comment

Experimental design

No comment

Validity of the findings

No comment

Additional comments

The revision is ready for publication.

Reviewer 4 ·

Basic reporting

No Comments

Experimental design

1. Authors have performed enzymatic degradation for the prepared hydrogel. However, for tissue engineering application perspective, authors need to perform hydrolytic and cellular degradation for a better understanding of the degradation profile of synthesized hydrogel. It could be performed based on the cited literature in your rebuttal letter. They are
[1] Mazzeo MS, et al. “Characterization of the Kinetics and Mechanism of Degradation of Human Mesenchymal Stem Cell-Laden Poly (ethylene glycol) Hydrogels.” ACS Appl Bio Mater 2019 2:81-92. 10.1021/acsabm.8b00390
[2] Yang G, et al. “Enzymatically crosslinked gelatin hydrogel promotes the proliferation of adipose tissue-derived stromal cells.” PeerJ 2016 4:e2497. 10.7717/peerj.2497
Discuss the degradation behaviour of the hydrogel in detail.

2. In one of the author's responses, the authors mentioned: "Observation and cytoskeletal staining showed that the cells grew and spread in the flat, and no obvious cell migration/penetration was observed". As the prepared hydrogels were intended for tissue engineering application, authors need to study the ADSCs survivability by embedding inside hydrogel materials.

Validity of the findings

No comments

---

## Round 0.3 · accepted · Accept

I appreciate all the hard work by the authors, the manuscript was improved by addressing the comments from the reviewers.

Reviewer 2 ·

Basic reporting

No Comments

Experimental design

No Comments

Validity of the findings

No Comments

Reviewer 4 ·

Basic reporting

The manuscript could be accepted for publication

Experimental design

Nil

Validity of the findings

Nil

Additional comments

Nil